# FAIR GENERATION VIA CONDITIONAL-INDEPENDENT FLOW MATCHING

## ABSTRACT

We study conditional prediction under group fairness constraints requiring the model's output $Y_{\text{pred}}$ to be conditionally independent of sensitive attributes $Z$ given the true label $Y_{\text{true}}$. Existing approaches—kernel penalties, adversarial debiasing, and mutual-information bounds—often scale poorly in high dimensions, are unstable to train, or lack auditable likelihoods. We propose **CI-CFM**, a conditional flow-matching generator coupled with density-based heads that score (i) the joint $p(Y_{\text{pred}}, Z, Y_{\text{true}})$ and (ii) a factorized reference $q(Y_{\text{pred}}, Z', Y_{\text{true}})$ together with their marginals $p(Z, Y_{\text{true}})$ and $q(Z', Y_{\text{true}})$. Our *divergence-difference* objective $\xi := D_{\text{KL}}(p\|q) - D_{\text{KL}}(p_{Z,Y_{\text{true}}}\|q_{Z',Y_{\text{true}}})$ *equals* the conditional mutual information $I(Y_{\text{pred}}; Z \mid Y_{\text{true}})$ under mild assumptions, enabling single-stage, non-adversarial training with tractable likelihoods for *auditing* conditional independence and efficient single-/few-step sampling at inference. On synthetic data and EHR benchmarks (MIMIC-III/IV), CI-CFM improves accuracy while substantially reducing dependence on $Z$ given $Y_{\text{true}}$; ablations confirm the effectiveness of the fairness weight schedule, variance-reduced estimator, and few-step ODE integration. Code is anonymously available at https://anonymous.4open.science/r/CICFM-0B67/.

## 1 INTRODUCTION

The fairness of modern generative models has been an increasing concern in the machine learning community. Fairness-aware methods aim to eliminate the bias inherent from the demographic groups (e.g., race or gender). Generative models, such as large language model chatbots, require not only high-fidelity samples, but also to comply with the fairness constraint. The fairness can be formulated as the following conditional independence (CI) constraint

$$Y_{\text{pred}} \perp Z \mid Y_{\text{true}},$$

where $Z$ denotes sensitive or nuisance attributes. Existing approaches typically (i) use kernel penalties (e.g., MMD/HSIC) that are bandwidth-sensitive, hard to tune, and scale poorly; (ii) rely on adversarial debiasing with unstable min–max optimization and brittle hyperparameter trade-offs; or (iii) pursue latent "disentanglement" with proxy supervision, which lacks principled density estimates and offers no direct way to measure conditional independence. As a result, auditing is difficult, training can be unstable, and accuracy often degrades—especially under irregular time series, structured missingness, and distribution shift common in EHR data. Meanwhile, (conditional) flow matching (FM/CFM) offers stable, simulation-free training with single-/few-step sampling, but prior work does not provide explicit CI control with auditable likelihoods. In this paper, we propose a generative framework based on FM, which enforces the aforementioned conditional independence constraint. Our contributions can be summarized as follows:

(1) We present a novel fairness-aware generative model based on conditional independence, which combines conditional flow matching with an explicit density regularizer to remove sensitive-attribute leakage while preserving task signal—using twin density heads and a divergence-difference loss for stable, non-adversarial training, tractable likelihood auditing, and efficient single-step sampling. (2) Theoretical analysis validates the appropriacy of our surrogate by proving that the divergence–difference $\xi$ is exactly equal to the conditional mutual information $I(Y_{\text{pred}}; Z \mid Y_{\text{true}})$, so minimizing $\xi$ is necessary and sufficient to enforce $Y_{\text{pred}} \perp Z, Y_{\text{true}}$ (Theorem 1). (3) Empirical

experiments on synthetic datasets and real-world datasets demonstrate the superior performance of our methods over existing competitors. Ablation analysis demonstrates the robustness to potential variations. (4) Qualitative studies on real-world datasets — MIMIC-III and MIMIC-IV demonstrate that our method alleviates the bias from sensitive attributes (e.g., ethnicity) when improving clinical predictions over the strongest non–ours baseline by *0.05–0.50* absolute points, while simultaneously reducing *EDDI* by *0.07–0.32*.

## 2 RELATED WORK

**Fairness-Aware Generation**    Post-processing and constrained learning enforce group fairness but may discard signal and limit test-time control (Hardt et al., 2016; Kleinberg et al., 2016; Woodworth et al., 2017). Representation approaches remove or obscure sensitive information (Zemel et al., 2013; Edwards & Storkey, 2015; Beutel et al., 2017; Zhang et al., 2018); adversarial variants align with domain-invariance (Ganin et al., 2016) but can be unstable, while variational/disentanglement methods improve sample efficiency at the cost of stronger latent assumptions and weaker CI guarantees (Louizos et al., 2015; Madras et al., 2018; Creager et al., 2019; Locatello et al., 2019). In healthcare/EHR, recent systems employ contrastive or counterfactual debiasing yet typically optimize surrogate discrepancies rather than directly enforcing CI (Rajkomar et al., 2018; Obermeyer et al., 2019; Wang et al., 2024; Oh et al., 2022; Liu et al., 2023).

**Enforcing CI.**    CI can be promoted via kernel criteria (e.g., HSIC, kernel CI tests) but these suffer from bandwidth sensitivity and scaling challenges in high dimensions (Gretton et al., 2005; Fukumizu et al., 2007; Zhang et al., 2011). Mutual-information penalties are conceptually appealing yet often rely on high-variance neural estimators and loose bounds (Barber & Agakov, 2004; Belghazi et al., 2018; Hjelm et al., 2019; Tschannen et al., 2020; Poole et al., 2019; McAllester & Stratos, 2020). Closer work trains CI regularizers alongside predictors (CIRCE), matches joint vs. factorized distributions adversarially (CI-GAN), or enforces CI in diffusion latents (CI-DiffAE), but these inherit kernel sensitivity or adversarial instability and add complexity via multiple losses and discriminators (Pogodin et al., 2023; Ahuja et al., 2021; Hwa et al., 2024). This motivates density-based objectives that scale and yield auditable likelihoods.

**Flows and flow matching.**    Normalizing flows enable exact likelihoods and efficient sampling via invertible architectures (Dinh et al., 2017; Rezende & Mohamed, 2015; Papamakarios et al., 2017; Kingma & Dhariwal, 2018). Continuous normalizing flows (neural ODEs) generalize flows to continuous-time dynamics, but require ODE solves during training and inference (Chen et al., 2018; Grathwohl et al., 2019). Flow matching (FM) replaces expensive integration with direct vector-field regression, yielding stable training and single-step sampling; rectified/stochastic interpolant variants further improve efficiency and sample quality (Lipman et al., 2023; Liu, 2022; Albergo et al., 2023). Building on this, (Tong et al., 2024) developed generalized conditional flow matching (CFM) to learn context-conditional generators. However, standard FM/CFM focuses on sample fidelity and training stability and does *not* provide explicit mechanisms for fairness or CI control. Our framework bridges this gap by pairing conditional flow matching with a density-based CI regularizer that trains joint and factorized likelihood surrogates in tandem.

**Our method's advantage**    Relative to adversarial CI enforcement (Zhang et al., 2018; Madras et al., 2018; Ahuja et al., 2021), our approach avoids discriminator dynamics, supplies explicit likelihood surrogates for joint and factorized models (enabling auditable CI), and scales to temporal inputs with a temporal encoder while preserving the stability and efficiency of flow matching (Lipman et al., 2023; Tong et al., 2024). Compared to kernel-based penalties (Gretton et al., 2005; Fukumizu et al., 2007; Pogodin et al., 2023), we eliminate bandwidth sensitivity and large-kernel costs via parametric density surrogates embedded in a divergence-difference objective aligned with conditional mutual information.

## 3 METHOD

We present our CI-CFM framework, where Figure 1 presents an overview. Our framework takes multivariate time series as input and first encodes them with a temporal encoder (e.g., transformer (Vaswani et al., 2017)) to produce a compact context vector. A conditional flow-matching generator then learns a vector field that transports a simple noise sample to the label space, and an ODE integrator produces

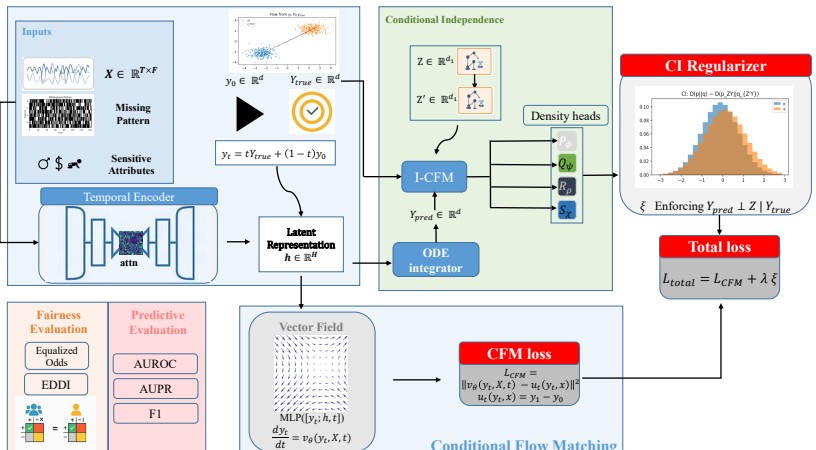

Figure 1: Our proposed conditional flow matching framework for fairness-aware generation.

the final prediction in one or a few steps. To enforce fairness, we attach a lightweight conditional independence module with shared backbone density heads that score both the joint distribution of (prediction, sensitive attributes) given the true label and a factorized counterpart built with an auxiliary sampler. Their divergence difference serves as an explicit conditional-independence regularizer that is added to the CFM objective. This design gives a single, non-adversarial training stage, stable optimization, tractable likelihoods for auditing, and efficient sampling—jointly improving predictive accuracy and reducing dependence on sensitive attributes. Algorithm 1 presents the detailed workflow of our method.

### 3.1 PROBLEM FORMULATION AND THEORETICAL MOTIVATION

Let $\mathcal{D} = \{(X_i, Y_{\text{true},i}, Z_i)\}_{i=1}^{N}$ be i.i.d. samples with $X_i \in \mathcal{X} \subset \mathbb{R}^{T \times F}$ (multivariate time series), $Y_{\text{true}} \in \mathcal{Y}$ (true target labels), and $Z \in \mathcal{Z}$ (a protected or sensitive attribute). Our objective is to learn a generator $G_\theta : \mathcal{X} \to \mathcal{Y}$ producing predictions $Y_{\text{pred}} = G_\theta(X)$ subject to the fairness constraint that, conditioned on the true label, the prediction should be statistically independent of the sensitive attribute:

$$Y_{\text{pred}} \perp Z \mid Y_{\text{true}}, \tag{1}$$

This conditional independence criterion ensures that prediction errors are not correlated with protected attributes within each true label class. It can also be rigorously quantified using conditional mutual information (CMI):

$$I\big(Y_{\text{pred}}; Z \mid Y_{\text{true}}\big) = \mathbb{E}\Big[\log \frac{p(Y_{\text{pred}}, Z \mid Y_{\text{true}})}{p(Y_{\text{pred}} \mid Y_{\text{true}})p(Z \mid Y_{\text{true}})}\Big],$$

which admits the KL-divergence form

$$I\big(Y_{\text{pred}}; Z \mid Y_{\text{true}}\big) = D_{\text{KL}}\big(p(Y_{\text{pred}}, Z \mid Y_{\text{true}})\|p(Y_{\text{pred}} \mid Y_{\text{true}})p(Z \mid Y_{\text{true}})\big).$$

where $D_{\text{KL}}$ denotes the Kullback-Leibler divergence. Since $D_{\text{KL}} \geq 0$ with equality if and only if the two distributions are identical almost surely, we have $I(Y_{\text{pred}}; Z \mid Y_{\text{true}}) \geq 0$ with equality if and only if the conditional independence holds. Thus, minimizing this quantity directly enforces our desired fairness constraint. However, estimating CMI in high-dimensional is challenging. Direct density estimation scales poorly with dimensionality and requires careful regularization to avoid overfitting.

To address these limitations, we propose a novel surrogate divergence measure that admits efficient estimation while preserving the theoretical guarantees of CMI. Following the theoretical framework of Ahuja et al. (2021), we introduce an auxiliary "factorized" distribution that decouples the prediction from the sensitive attribute:

$$q(Y_{\text{pred}}, Z', Y_{\text{true}}) = p(Y_{\text{pred}}, Y_{\text{true}})q(Z' \mid Y_{\text{true}}), \tag{2}$$

where $Z'$ denotes the an independent draw from the conditional distribution $q(Z' \mid Y_{\text{true}})$. In practice, $Z'$ is sampled from the empirical label-conditional distribution via within-class permutation of $Z$ (with optional Gaussian), preserving $p(Z \mid Y_{\text{true}})$ while breaking its dependence on $Y_{\text{pred}}$. The estimator $\xi$ is invariant to the choice of $q$(see Appendix A.1.3 for the detail). This ensures the marginal distribution between predictions and labels is maintained, without spurious correlations to sensitive attributes. We then define our surrogate divergence measure as:

$$\xi = D_{\text{KL}}\left(p(Y_{\text{pred}}, Z, Y_{\text{true}}) \parallel q(Y_{\text{pred}}, Z', Y_{\text{true}})\right) - D_{\text{KL}}\left(p(Z, Y_{\text{true}}) \parallel q(Z', Y_{\text{true}})\right) \quad (3)$$

A large body of work regularizes representations by *bounding* or *estimating* mutual information with variational lower bounds or contrastive estimators (e.g., MINE, Deep InfoMax, and related contrastive/InfoNCE objectives) (Belghazi et al., 2018; Hjelm et al., 2019; Poole et al., 2019; Tschannen et al., 2020). These approaches can introduce non-negligible bias/variance trade-offs and instability in high dimensions. In contrast, our *divergence-difference* construction couples joint and marginal KLs built from label-preserving resampled references (2). The following theorem shows that this surrogate *exactly equals* the CMI:

**Theorem 1** (Equivalence of $\xi$ and Conditional Mutual Information)**.** *Under the mild regularity conditions in Appendix A.1.1—namely, absolute continuity and strict positivity of the relevant densities, a factorized reference constructed via label-wise resampling, and use of the strictly convex, separable KL divergence—the surrogate $\xi$ coincides exactly with the CMI:*

$$\xi \;=\; I\big(Y_{\text{pred}}; Z \mid Y_{\text{true}}\big).$$

*In particular,*

$$\xi \geq 0, \qquad \xi = 0 \iff Y_{\text{pred}} \perp Z \,\big|\, Y_{\text{true}}.$$

*Proof.* See Appendix A.1.

**Remark.** The identity $\xi = I(Y_{\text{pred}}; Z \mid Y_{\text{true}})$ is *invariant* to the specific choice of the auxiliary kernel $q(Z' \mid Y_{\text{true}})$ as long as standard regularity holds: $q(\cdot \mid Y_{\text{true}})$ is absolutely continuous on the support of $p(Z \mid Y_{\text{true}})$, and the involved KL terms are finite. Any such Markov kernel yields the same value of $\xi$, hence the same CMI. Our practical instantiation—within-label permutation of $Z$ with a small Gaussian perturbation to ensure positivity—is merely convenient rather than essential (see Appendix A.1.3).

## 3.2 Conditional Flow Matching for Fair Generation

To implement our fairness-aware generator $G_\theta(X)$, we leverage the recently developed conditional flow matching (CFM) framework (Lipman et al., 2023), which provides a simulation-free approach to training continuous normalizing flows (CNFs) (Grathwohl et al., 2019).

CNFs define the diffeomorphisms $\{\phi_t\}_{t \in [0,1]}$ through the ordinary differential equation(ODE):

$$\frac{d\phi_t(x)}{dt} = v_t\big(\phi_t(x)\big), \quad \phi_0(x) = x,$$

where $v_t : \mathbb{R}^d \to \mathbb{R}^d$ is a time-dependent vector field. The flow $\phi_t$ transforms a simple base distribution $p_0$ (typically standard Gaussian) into a target distribution $p_1$ via the push-forward operation:

$$p_t = [\phi_t]_*(p_0) = p_0\big(\phi_t^{-1}(x)\big) \left|\det \frac{\partial \phi_t^{-1}(x)}{\partial x}\right|.$$

Traditional training of CNFs via maximum likelihood requires solving the flow equations during both forward and backward passes, leading to significant computational overhead and potential numerical instabilities (Chen et al., 2018; Grathwohl et al., 2019). Flow matching circumvents these issues by directly regressing the vector field against a known target velocity field $u_t(x)$ that generates the desired probability path $\{p_t\}_{t \in [0,1]}$ $(\partial_t p_t + \nabla \cdot (p_t u_t) = 0)$.

$$\mathcal{L}_{\text{FM}}(\theta) = \mathbb{E}_{t \sim \text{U}[0,1], x \sim p_t}\left[\left\|v_t(x; \theta) - u_t(x)\right\|^2\right].$$

However, computing the marginal target field $u_t(x)$ is generally intractable. CFM resolves this tractability issue by exploiting a mixture-of-paths decomposition. Suppose the marginal probability path admits the representation:

$$p_t(x) \;=\; \int p_t(x \mid z)\, q(z)\, \mathrm{d}z,$$

where $z$ is a conditioning variable sampled from $q(z)$. If each conditional path $p_t(x \mid z)$ is generated by a (tractable) vector field $u_t(x \mid z)$ satisfying $\partial_t p_t(x \mid z) + \nabla \cdot \big(p_t(x \mid z)\, u_t(x \mid z)\big) = 0$, then the *marginal* velocity field $u_t(x) := \mathbb{E}_{z \sim q(z)}\Big[ u_t(x \mid z)\, \frac{p_t(x \mid z)}{p_t(x)} \Big]$ also generates the marginal path $p_t(x)$. Direct regression to $u_t(x)$ is intractable since it requires evaluating the integral above. Instead, CFM minimizes the *conditional* regression objective

$$\mathcal{L}_{\mathrm{CFM}}(\theta) := \mathbb{E}_{\substack{t \sim \mathrm{Unif}[0,1] \\ z \sim q(z) \\ x \sim p_t(\cdot \mid z)}} \big\| v_t(x; \theta) \,-\, u_t(x \mid z) \big\|^2,$$

where $v_t(x; \theta)$ is a neural network approximation of the vector field. Under the mild positivity condition $p_t(x) > 0$ for all $x, t$, one can show (Theorem 3.2) that, up to an additive constant, $\mathcal{L}_{\mathrm{CFM}}(\theta) = \mathcal{L}_{\mathrm{FM}}(\theta)$ and hence $\nabla_\theta \mathcal{L}_{\mathrm{CFM}} = \nabla_\theta \mathcal{L}_{\mathrm{FM}}$. In practice, CFM thus recovers the same optimal dynamics as FM while requiring only samples $(z, x) \sim q(z)\, p_t(x \mid z)$ and evaluations of $u_t(x \mid z)$, thereby circumventing the need to compute or estimate the intractable marginal velocity $u_t(x)$.

## 3.3 Model Architecture

In this work, we utilize I-CFM (Independent Coupling CFM) (Tong et al., 2024), which identifies the conditioning variable $z$ with a source-target pair $(y_0, y_1)$ where $y_0 \sim \mathcal{N}(0, I)$ is a source point and $y_1 = Y_{\mathrm{true}}$ is the target. Use the linear interpolant $y_t = (1 - t)y_0 + ty_1$ and constant velocity $u_t(y_t \mid y_0, y_1) = y_1 - y_0$. Given covariates $X$, the generator integrates the learned vector field to produce the prediction

$$Y_{\mathrm{pred}} = G_\theta(X) := \phi_1^\theta(y_0; X), \qquad \frac{d\phi_t^\theta}{dt}(y_0; X) = v_\theta\big(\phi_t^\theta(y_0; X), X, t\big), \phi_0^\theta(y_0; X) = y_0.$$

This construction ensures that the learned flow transports samples from the base distribution directly to the target predictions along straight-line paths, providing both computational efficiency and theoretical guarantees.

For inputs $X \in \mathbb{R}^{T \times F}$, we employ a general temporal encoder that captures multi-scale temporal dependencies,

$$h = \mathrm{Enc}(X) \in \mathbb{R}^H.$$

Formally, $\mathrm{Enc} : \mathbb{R}^{T \times F} \to \mathbb{R}^H$ is a learnable function that maps the time-series $X$ to a fixed-length contextual representation $h$. The generator parameterizes a time-dependent vector field by conditioning on this context:

$$v_\theta(y_t, X, t) = \mathrm{MLP}\big([\, y_t;\, h;\, t\,]\big),$$

where $h$ provides the temporal summary of $X$ and $[\cdot; \cdot]$ denotes concatenation.

The final generator produces samples by numerically integrating the learned ODE:

$$\frac{dy}{dt} = v_\theta(y_t, X, t), \quad y_0 \sim \mathcal{N}(0, I)$$

from $t = 0$ to $t = 1$ using integration schemes with $S$ steps (step size $\Delta t = 1/S$). (e.g., Heun/RK2 or Euler).

## 3.4 Fairness-Constrained Training Framework

The primary training objective follows the CFM paradigm, utilizing the loss given by:

$$\mathcal{L}_{\mathrm{CFM}}(\theta) = \mathbb{E}_{t, X, Y_{\mathrm{true}}, y_0} \big[ \|v_\theta(y_t, X, t) - u_t(y_t, X)\|^2 \big]$$

where the expectation is taken over $t \sim \mathrm{Uniform}[0, 1]$, training pairs $(X, Y_{\mathrm{true}})$, and base samples $y_0 \sim \mathcal{N}(0, I)$.

To compute the surrogate divergence $\xi$ in (3), we estimate both KL terms via four neural density estimators trained with CFM framework: (i) $P_\phi$ for $\log p(Y_{\text{pred}}, Z, Y_{\text{true}})$, (ii) $Q_\psi$ for $\log q(Y_{\text{pred}}, Z', Y_{\text{true}})$ with $Z' \perp Y_{\text{pred}} \mid Y_{\text{true}}$, (iii) $R_\rho$ for $\log p(Z, Y_{\text{true}})$, and (iv) $S_\chi$ for $\log q(Z', Y_{\text{true}})$.

Once trained, each flow provides log-density estimates by the continuous change-of-variables identity:

$$\log p_T(\zeta) = \log p_0(\zeta_0) - \int_0^T \nabla_\zeta \cdot v(\phi_{t\leftarrow 0}(\zeta_0), t)\, dt, \qquad \zeta_0 = \phi_{0\leftarrow T}(\zeta),$$

where $\zeta$ denotes the variables in the respective flow model, and divergence is computed by automatic differentiation or stochastic trace estimators. This yields $\log P_\phi, \log Q_\psi, \log R_\rho, \log S_\chi$ consistently across the four heads.

To instantiate the factorized $q(\cdot)$, we draw $Z'$ by label-wise permutation plus small Gaussian noise:

$$Z'^{(i)} = Z^{(\sigma_{Y_{\text{true}}^{(i)}}(i))} + \epsilon^{(i)}, \qquad \epsilon^{(i)} \sim \mathcal{N}(0, \sigma_{\text{noise}}^2 I),$$

where $\sigma_y$ is a random permutation of the sample indices that is specific to the value of the label $y = Y_{\text{true}}^{(i)}$. This operation preserves the conditional marginal $p(Z \mid Y_{\text{true}})$ while breaking the dependence on $Y_{\text{pred}}$.

Given a minibatch $\{(Y_{\text{pred}}^{(i)}, Z^{(i)}, Z'^{(i)}, Y_{\text{true}}^{(i)})\}_{i=1}^B$, we form the per-sample joint and marginal log-density differences

$$
\begin{aligned}
\delta_{\text{joint}}^{(i)} &= \log P_\phi(Y_{\text{pred}}^{(i)}, Z^{(i)}, Y_{\text{true}}^{(i)}) - \log Q_\psi(Y_{\text{pred}}^{(i)}, Z'^{(i)}, Y_{\text{true}}^{(i)}), \\
\delta_{\text{marg}}^{(i)} &= \log R_\rho(Z^{(i)}, Y_{\text{true}}^{(i)}) - \log S_\chi(Z'^{(i)}, Y_{\text{true}}^{(i)}).
\end{aligned}
\tag{4}
$$

We then apply exponential moving-average (EMA) baselines to reduce variance:

$$b_{\text{joint}} \leftarrow \beta b_{\text{joint}} + (1-\beta)\frac{1}{B}\sum_i \delta_{\text{joint}}^{(i)}, \qquad b_{\text{marg}} \leftarrow \beta b_{\text{marg}} + (1-\beta)\frac{1}{B}\sum_i \delta_{\text{marg}}^{(i)},$$

and define the variance-reduced Monte Carlo estimator

$$\widehat{\xi} = \frac{1}{B}\sum_{i=1}^B \left[(\delta_{\text{joint}}^{(i)} - b_{\text{joint}}) - (\delta_{\text{marg}}^{(i)} - b_{\text{marg}})\right].$$

Only the *joint* difference depends on the generator output $Y_{\text{pred}}$ and hence on $\theta$. The marginal term involves only $(Z, Y_{\text{true}})$ or $(Z', Y_{\text{true}})$ and carries **no** path to $\theta$. We therefore backpropagate through $\delta_{\text{joint}}$ to update $(\theta, \phi, \psi)$, and *stop the gradient* from $\delta_{\text{marg}}$ to $\theta$ (while using $\delta_{\text{marg}}$ to update $(\rho, \chi)$).

The overall training objective combines the CFM loss for the generator with the fairness surrogate:

$$\mathcal{L}_{\text{total}}(\theta, \phi, \psi, \rho, \chi) = \mathcal{L}_{\text{CFM}}(\theta) + \lambda\,\widehat{\xi}(\theta, \phi, \psi, \rho, \chi), \qquad \lambda > 0.$$

A short warm-up phase for $\lambda$ is applied to enhance training stability in initial iterations.

This integrated training framework ensures the generator produces accurate predictions while satisfying conditional independence constraints, offering a principled approach to fairness-aware time series generation. Complete pseudocode (Algorithm 1) is provided in Appendix A.2.

**Convergence and Complexity Analysis.** We conclude the section by summarizing the convergence and complexity properties of the proposed CI–CFM framework. Under standard regularity assumptions (stated in the appendix A.1.1), the CFM objective $\mathcal{L}_{\text{CFM}}(\theta)$ converges at rate $\mathcal{O}(B^{-1/2})$ following standard flow matching analysis, while our fairness regularizer $\widehat{\xi}$ achieves the same convergence rate to the true conditional mutual information $I(Y_{\text{pred}}; Z|Y_{\text{true}})$ via consistent neural density estimation and EMA variance reduction. Compared to vanilla CFM with complexity $\mathcal{O}(BS \cdot \dim(Y))$, our method incurs $\mathcal{O}(BS \cdot d_{\max} + B \cdot C_{\text{fair}})$ per-iteration cost, where

Table 1: **Synthetic Regression Performance.** $\text{Mean}_{(\min,\max)}$ over 10 seeds. The CI regularizer substantially tightens the three CI diagnostics with only marginal changes in RMSE/MAE.

| Method | RMSE ($\downarrow$) | MAE ($\downarrow$) | $\text{EDDI}_{\text{cont}}$ ($\downarrow$) | $|\rho|_{\text{partial}}$ ($\downarrow$) | $\text{HSIC}_{\text{partial}}$ ($\downarrow$) |
|---|---|---|---|---|---|
| CFM ($\lambda=0$) | $1.12_{(1.12,\,1.13)}$ | $0.923_{(0.918,\,0.927)}$ | $0.0467_{(0.0396,\,0.0520)}$ | $0.136_{(0.126,\,0.145)}$ | $0.000859_{(0.000673,\,0.00102)}$ |
| CFM+CI ($\lambda=0.1$) | $1.12_{(1.11,\,1.12)}$ | $\mathbf{0.900}_{(0.894,\,0.907)}$ | $\mathbf{0.0345}_{(0.0293,\,0.0403)}$ | $\mathbf{0.121}_{(0.111,\,0.138)}$ | $\mathbf{0.000478}_{(0.000369,\,0.000567)}$ |

$d_{\max} = \max\{\dim(Y) + \dim(Z) + \dim(Y_{\text{true}}), \dim(Z) + \dim(Y_{\text{true}})\}$ and $C_{\text{fair}}$ captures fairness-specific overhead. This translates to approximately 3–5× computational increase and 2–3× memory overhead, primarily due to four density heads and gradient routing operations. The warm-up schedule with $\lambda$ increasing from 0 to $\lambda_{\max}$ over $E_{\text{warm}}$ epochs ensures stable convergence to a stationary point of the combined objective $\mathcal{L}_{\text{total}}$ at rate $\mathcal{O}(T^{-1/2})$ for $T$ total iterations under standard SGD analysis, while the gradient routing strategy prevents conflicting signals between CFM and fairness objectives. Unlike diffusion-based debiasing methods requiring iterative sampling or adversarial approaches with min-max instabilities, our framework provides explicit fairness control with single forward-pass generation, trading increased training cost for superior inference efficiency and theoretical interpretability.

## 4 SIMULATION STUDIES

### 4.1 EXPERIMENTAL SETUP

We synthesize $n$ i.i.d. triples $\{(X^{(i)}, Y^{(i)}, Z^{(i)})\}_{i=1}^n$ together with a binary group label $G^{(i)} \in \{0, 1\}$ (extension to $K > 2$ is straightforward). For each $i$, the raw multivariate time series $X^{(i)} \in \mathbb{R}^{T \times F}$ has entries

$$X_{t,j}^{(i)} = \big(0.5 + \eta_{i,j}\big) \sin(\omega_j t + \phi_{i,j}) + 0.1\, \varepsilon_{t,j}^{(i)}, \quad \eta_{i,j} \sim \mathcal{N}(0, 0.2^2),\ \phi_{i,j} \sim \mathcal{N}(0, \pi^2),\ \varepsilon_{t,j}^{(i)} \sim \mathcal{N}(0, 1), \tag{5}$$

with piecewise frequencies

$$\omega_j = \begin{cases} 0.2 + 0.1(j - 1), & 1 \le j \le \lfloor F/3 \rfloor, \\ 1.0 + 0.5\big(j - 1 - \lfloor F/3 \rfloor\big), & \lfloor F/3 \rfloor < j \le \lfloor 2F/3 \rfloor, \\ 3.0 + 1.0\big(j - 1 - \lfloor 2F/3 \rfloor\big), & \lfloor 2F/3 \rfloor < j \le F. \end{cases}$$

We compute the discrete Fourier transform along the temporal axis and keep the lowest $T/4$ modes, $\widehat{X}^{(i)} = |\text{FFT}(X^{(i)}, \dim=1)|$, and form $\mathbf{x}_i^{(s)} = \frac{1}{T/4} \sum_{k=1}^{T/4} \widehat{X}_{k,:}^{(i)}$, $\mathbf{x}_i^{(m)} = \frac{1}{T} \sum_{t=1}^T X_{t,:}^{(i)}$, and $\mathbf{x}_i^{(\sigma)} = \sqrt{\frac{1}{T} \sum_{t=1}^T (X_{t,:}^{(i)} - \mathbf{x}_i^{(m)})^2}$. The final feature vector is $\mathbf{h}_i = [\mathbf{x}_i^{(s)}; \mathbf{x}_i^{(m)}; \mathbf{x}_i^{(\sigma)}] \in \mathbb{R}^{3F}$.

Let $s(G) \in \{-1, +1\}$ be the centered encoding of $G$. Draw $W_Y \sim \mathcal{N}(0, 0.3^2 I_{3F})$ and define a baseline $Y_{\text{base}}^{(i)} = \tanh(\mathbf{h}_i W_Y) + \varepsilon_i^Y$, with $\varepsilon_i^Y \sim \mathcal{N}(0, 0.05^2 I_{F_1})$. We inject two optional group effects:

$$Y^{(i)} \leftarrow \big(1 + \kappa_Y\, s(G^{(i)})\big) \odot \Big(Y_{\text{base}}^{(i)} + \gamma_Y\, s(G^{(i)})\Big), \tag{6}$$

where $\gamma_Y$ controls an additive shift and $\kappa_Y$ a multiplicative scale. Next draw $W_Z \sim \mathcal{N}(0, 0.4^2 I_{F_1})$ and generate

$$Z^{(i)} = \alpha_Y\big(Y^{(i)} W_Z\big) + \gamma_Z\, s(G^{(i)})\, \mathbf{v} + \varepsilon_i^Z, \qquad \varepsilon_i^Z \sim \mathcal{N}(0, 0.08^2 I_{F_2}), \tag{7}$$

where $\alpha_Y$ controls the $Y \to Z$ coupling, $\gamma_Z$ tunes inter-group separability in $Z$, and $\mathbf{v} \in \mathbb{R}^{F_2}$ is a random direction.

All random draws are performed with a fixed seed for reproducibility. We report results averaged over multiple independent replicates. Full simulation settings—including data generation, model architectures, training/testing protocol, and metric computation—are provided in Appendix B.

### 4.2 RESULTS.

We evaluate predictive accuracy (RMSE, MAE) and conditional independence using three diagnostics—$\text{EDDI}_{\text{cont}}$, partial correlation, and residual HSIC—whose precise definitions, preprocessing (binning, residualization), and kernel/bandwidth choices are deferred to Appendix B.1.

Table 1 reports mean$_{(min,max)}$ over 10 seeds: relative to plain CFM ($\lambda=0$), CFM+CI ($\lambda=0.1$) reduces EDDI$_{cont}$ from 0.0467 to 0.0345 ($\sim 26\%$), the average absolute partial correlation from 0.136 to 0.121 ($\sim 11\%$), and residual HSIC from $8.59 \times 10^{-4}$ to $4.78 \times 10^{-4}$ ($\sim 44\%$), while RMSE/MAE vary by at most $\approx$1–2%. The tight min–max intervals across seeds indicate that these CI gains are consistent and robust. Figure 2 corroborates this trend by plotting the Pareto curve of $-$RMSE versus EDDI$_{cont}$ as $\lambda$ varies: increasing $\lambda$ consistently shifts solutions *leftward* (smaller EDDI$_{cont}$, hence stronger CI) with only minor vertical movement (RMSE), yielding a smooth frontier with a clear "elbow" around $\lambda \approx 0.1$. Together, these results indicate that substantial CI gains—now measured by EDDI$_{cont}$—are attainable with negligible impact on predictive accuracy, and are stable across random seeds.

## 5 REAL-DATA EVALUATION

### 5.1 EXPERIMENT SETUP

**Datasets.** We evaluate on two large-scale critical-care EHR corpora: *MIMIC-III* (Johnson et al., 2016) and *MIMIC-IV* (Johnson et al., 2023). Both consist of de-identified records from ICU or emergency-department stays at BIDMC, including multivariate time series (vitals, labs) and demographics. Following common preprocessing in clinical time-series benchmarks (Harutyunyan et al., 2019; Wang et al., 2024), we extract hourly measurements from the first 48 hours after ICU admission (26 variables for MIMIC-III and 25 for MIMIC-IV) and use standard train/validation/test splits. Sensitive attributes used for fairness auditing include insurance, marital status, race, gender, and age (Wang et al., 2024).

**Tasks & Evaluation Metrics.** Consistent with prior clinical prediction work (Harutyunyan et al., 2019; Wang et al., 2024), we study two binary outcomes: (1) **In-hospital mortality (IHM)**—whether the patient dies during the index hospitalization; and (2) **30-day readmission (READM)**—whether the patient is readmitted within 30 days of discharge. We report AUROC, AUPR, and F1 for predictive performance, and adopt Equalized Odds (EO) and the Error Distribution Disparity Index (EDDI) as group-fairness metrics (Wang et al., 2024). All metrics are accompanied by 95% bootstrap confidence intervals.

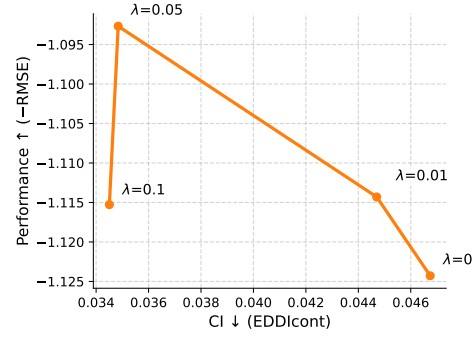

Figure 2: **Performance–fairness trade-off.** Pareto curve of performance ($-$ RMSE, higher is better) versus CI (here EDDI$_{cont}$, lower is better) as the regularization weight $\lambda$ varies. Larger $\lambda$ moves solutions leftward (weaker dependence on $Z$ given $Y$) with only minor vertical movement, illustrating that CI can be improved with limited impact on predictive accuracy.

**Comparable Methods.** (1) Backbone only without constraints on conditional independence: Transformer (Vaswani et al., 2017), LSTM (Graves & Graves, 2012), RNN (Elman, 1990), and CNN (LeCun et al., 1998); (2) Fairness-aware models: Including general fairness models: FFVAE (Creager et al., 2019), FarconVAE (Oh et al., 2022) and clinical models specifically on EHRs: FairEHR-CLP (Wang et al., 2024), FLMD (Liu et al., 2023). See Appendix C for implementation details and baseline configurations of all comparable methods.

### 5.2 EXPERIMENT RESULTS

**Quantitative Results.** Table 2 reports results on MIMIC-III and MIMIC-IV for in-hospital mortality (IHM) and readmission (READM) under a single training stage. Our CI-CFM consistently surpasses traditional EHR predictors (e.g., LSTM, RNN, CNN, Transformer) and fairness-aware baselines (e.g., FairEHR-CLP, FLMD, FFVAE, FarconVAE) on classification metrics (AUROC, AUPR, F1) and on fairness metrics (EO, EDDI). Notably, CI-CFM attains the highest *AUROC* across all four evaluations, improving over the strongest non–ours baseline by *0.05–0.50* absolute points, while simultaneously reducing *EDDI* by *0.07–0.32*. These gains come from (i) a flow-matching generator that captures long-range temporal structure and irregular sampling without adversarial training, and (ii) an explicit, density-based CI regularizer that suppresses residual dependence on sensitive attributes yet preserves label-discriminative signal.

Table 2: Performance comparison on MIMIC-III and MIMIC-IV for In-Hospital Mortality and Readmission tasks. All results(%) are reported with 95% confidence intervals. The best results are highlighted in **bold**, and the second-best results are underlined. Avg. Rank indicates the average ranking of each method across five evaluation metrics.

| Model | In-Hospital Mortality | | | | | | Readmission | | | | | |
|---|---|---|---|---|---|---|---|---|---|---|---|---|
| | AUROC (↑) | AUPR (↑) | F1 (↑) | EO (↓) | EDDI (↓) | Avg. Rank | AUROC (↑) | AUPR (↑) | F1 (↑) | EO (↓) | EDDI (↓) | Avg. Rank |
| Dataset 1: MIMIC-III | | | | | | | | | | | | |
| Transformer Vaswani et al. (2017) | 80.49(78.58, 82.48) | 38.54(33.95, 43.61) | 41.09(37.64, 44.56) | 10.09(8.77, 14.54) | 4.13(3.56, 5.91) | 7.6 | 70.26(67.85, 72.49) | 38.87(35.09, 43.15) | 40.43(37.51, 43.36) | 8.97(8.55, 13.51) | 6.05(4.67, 8.68) | 7.8 |
| LSTM (Graves & Graves, 2012) | 82.41(80.55, 84.38) | 42.34(37.66, 47.69) | 43.76(40.56, 47.19) | 8.77(8.53, 13.81) | 4.76(3.98, 6.46) | 5.0 | 72.39(70.17, 74.71) | 39.35(35.67, 43.51) | 41.58(38.84, 44.5) | 8.12(7.97, 12.95) | 6.15(4.62, 8.57) | 6.0 |
| RNN Elman (1990) | 81.39(79.31, 83.50) | 43.38(38.45, 48.06) | 43.26(39.75, 46.98) | 8.32(8.08, 13.56) | 4.06(3.47, 6.02) | 4.4 | 71.60(69.41, 73.81) | 38.93(35.15, 42.91) | 43.18(40.42, 46.00) | 9.46(8.32, 13.75) | 4.63(3.98, 7.57) | 7.0 |
| CNN (LeCun et al., 1998) | 82.28(80.39, 84.17) | **44.19**(39.37, 49.07) | 44.03(40.35, 47.47) | 8.48(8.33, 13.19) | 4.58(3.69, 6.11) | 4.0 | 74.15(72.14, 76.30) | 42.55(38.50, 46.78) | 43.18(40.42, 46.00) | 9.46(8.32, 13.75) | 5.57(4.32, 8.29) | 3.2 |
| FairEHR-CLP (Wang et al., 2024) | 79.70(77.63, 81.86) | 35.83(31.61, 40.29) | 40.35(36.79, 43.75) | 8.46(8.30, 13.63) | 4.32(3.46, 6.31) | 7.7 | 73.72(71.41, 75.98) | 38.96(35.05, 43.30) | 41.65(39.09, 43.97) | 8.59(8.28, 14.11) | 6.10(4.26, 9.32) | 5.4 |
| FLMD (Liu et al., 2023) | 81.77(79.74, 83.72) | 41.72(37.13, 46.91) | 43.51(39.87, 47.12) | 9.73(8.95, 14.54) | 4.32(3.80, 5.83) | 5.9 | 73.27(71.13, 75.48) | 41.08(37.26, 45.40) | 42.18(39.41, 44.84) | 7.87(7.28, 13.26) | 5.72(4.53, 8.17) | 2.6 |
| FFVAE (Creager et al., 2019) | 82.19(80.27, 84.06) | 41.06(36.23, 45.85) | 41.73(38.08, 45.26) | 7.86(6.32, 11.08) | 3.35(2.84, 5.42) | 4.4 | 71.84(69.48, 73.96) | 37.50(33.83, 41.71) | 40.57(37.92, 42.95) | 8.05(7.56, 12.98) | 4.53(3.47, 7.37) | 5.4 |
| FarconVAE (Oh et al., 2022) | 82.27(80.21, 84.10) | 40.80(35.96, 45.81) | 42.93(39.17, 46.56) | 7.38(5.10, 15.42) | 2.40(1.50, 4.05) | 4.4 | 73.70(71.52, 75.83) | 39.56(35.80, 43.81) | 42.05(38.91, 45.14) | 8.99(6.11, 16.04) | 3.82(2.48, 7.00) | 3.4 |
| CI-CFM(Ours) | **82.73**(80.60, 84.70) | 43.33(38.60, 48.41) | **44.15**(40.73, 47.87) | **7.31**(6.99, 11.87) | **2.22**(2.09, 4.86) | **1.6** | **74.20**(71.99, 76.41) | **42.57**(38.74, 46.52) | **44.15**(41.38, 46.77) | **6.43**(6.5, 11.54) | **3.5**(3.17, 6.61) | **1.0** |
| Dataset 2: MIMIC-IV | | | | | | | | | | | | |
| Transformer Vaswani et al. (2017) | 82.35(80.61, 84.16) | 40.98(36.80, 45.71) | 42.94(39.56, 46.23) | 5.11(4.73, 9.05) | 3.02(2.54, 4.22) | 7.0 | 71.96(69.92, 73.95) | 39.87(36.55, 43.65) | 42.05(39.22, 45.00) | 6.20(5.43, 8.99) | 3.20(2.70, 5.01) | 7.8 |
| LSTM (Graves & Graves, 2012) | 82.93(81.08, 84.83) | 45.56(40.89, 49.95) | 43.50(40.08, 46.60) | 5.27(5.22, 8.55) | 3.20(2.71, 4.26) | 5.6 | 73.09(71.10, 75.00) | 43.63(39.94, 46.85) | 42.98(40.35, 45.39) | 7.52(6.18, 10.63) | 3.69(3.05, 5.66) | 6.0 |
| RNN Elman (1990) | 82.39(80.46, 84.08) | 43.65(39.37, 48.06) | 44.18(40.52, 47.79) | 5.69(4.88, 9.29) | 2.53(2.03, 3.52) | 6.0 | 72.66(70.75, 74.60) | 42.32(38.75, 45.87) | 43.20(40.88, 45.65) | 6.00(5.62, 9.33) | 3.78(2.94, 5.96) | 6.2 |
| CNN (LeCun et al., 1998) | 83.56(81.66, 85.27) | **46.65**(42.02, 51.09) | 44.57(40.93, 48.03) | 5.18(5.30, 8.37) | 3.03(2.50, 4.02) | 4.0 | 73.74(71.89, 75.66) | 42.46(38.93, 45.94) | 44.07(41.29, 46.66) | 7.26(6.45, 10.41) | 3.81(2.91, 5.75) | 5.1 |
| FairEHR-CLP (Wang et al., 2024) | 83.45(81.65, 85.14) | 42.07(37.63, 46.57) | 40.48(36.38, 44.17) | 5.84(5.20, 9.54) | 2.35(2.12, 3.53) | 6.4 | 74.36(72.39, 76.32) | 41.62(37.93, 45.08) | 42.60(39.73, 45.40) | 7.27(6.20, 9.90) | 3.11(2.46, 4.85) | 6.0 |
| FLMD (Liu et al., 2023) | 83.69(81.78, 85.50) | 44.80(40.16, 49.45) | 43.44(39.47, 47.33) | 3.63(3.46, 7.39) | 2.36(1.92, 3.37) | 3.4 | 74.24(72.23, 76.14) | 42.43(38.68, 46.25) | 42.87(39.83, 46.03) | 4.03(3.75, 7.01) | 2.69(2.25, 4.22) | 3.8 |
| FFVAE (Creager et al., 2019) | 82.46(80.54, 84.28) | 41.44(36.72, 46.22) | 42.74(38.59, 46.48) | 4.55(4.05, 8.02) | 2.15(1.70, 3.18) | 5.4 | 73.20(71.16, 75.14) | 41.99(36.45, 45.26) | 42.37(39.61, 45.26) | 4.47(3.96, 7.23) | 2.70(2.36, 4.45) | 5.6 |
| FarconVAE (Oh et al., 2022) | 83.14(81.30, 84.88) | 40.30(35.66, 45.10) | 42.54(38.74, 45.86) | 4.78(2.92, 7.76) | 1.74(1.24, 2.69) | 5.6 | 73.42(71.46, 75.37) | 42.47(38.99, 46.01) | 44.01(41.21, 47.08) | 4.87(3.15, 7.51) | 2.40(1.54, 3.47) | 3.4 |
| CI-CFM(Ours) | **83.88**(82.15, 85.50) | 44.13(39.46, 48.90) | **45.33**(40.14, 47.52) | **3.49**(3.17, 5.01) | **1.66**(1.49, 2.88) | **1.6** | **74.86**(73.03, 76.81) | **43.72**(40.24, 47.48) | **44.72**(41.78, 47.65) | **3.98**(3.66, 7.13) | **2.33**(2.13, 4.07) | **1.0** |

**Qualitative Results.** Figure 3 visualizes the equalized odds across different sensitive attributes (Insurance, Marital, Race, Gender, Age). In both tasks the polygon for CI-CFM (Ours) is uniformly contained within those of all baselines, yielding the smallest enclosed area (i.e., lowest aggregate EO). The largest gains appear on *Gender* and *Marital*, where our method produces visibly shorter radii—often several points lower than FairEHR-CLP and FLMD—and it remains competitive on *Insurance*. *Race* is the most challenging attribute for every method (all curves peak on the Race axis), yet our CI-regularized model consistently attenuates this spike relative to FFVAE/FarconVAE.

The **READM** panel exhibits the same pattern as **IHM**, indicating stability across targets: EO decreases monotonically for most axes, with particularly pronounced reductions on Gender and Marital and moderate but consistent improvements on Age. The results demonstrate that our proposed CI-CFM consistently achieves superior fairness (lower EO values) across all sensitive attributes compared to baseline methods.

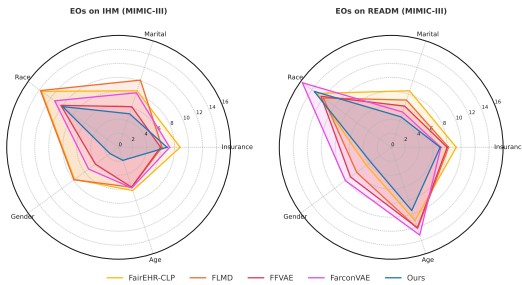

Figure 3: The Equalized Odds of CI-CFM on sensitive attributes on MIMIC-III dataset.

# 6 CONCLUSION

We introduced *CI–CFM*, a fairness-aware conditional generator that integrates conditional flow matching with a density-based conditional-independence regularizer. Under mild assumptions (Theorem 1), our divergence–difference objective is exactly equal to the conditional mutual information $I(Y_{\text{pred}}; Z \mid Y_{\text{true}})$, enabling single-stage, non-adversarial training with auditable likelihoods and efficient few-step sampling. On both synthetic data and MIMIC-III/IV, CI–CFM improves predictive accuracy while markedly reducing dependence on sensitive attributes (EO/EDDI), achieving state-of-the-art performance in accuracy–fairness trade-offs.

**Limitations & future work.** CI–CFM employs learned "density" heads; model misspecification can bias the CI surrogate. Our factorized reference is instantiated via within-label permutation with small noise, which may be suboptimal under label imbalance or label shift. The method targets a specific constraint $Y_{\text{pred}} \perp Z \mid Y_{\text{true}}$ and introduces moderate training overhead relative to vanilla CFM. Future work includes amortized or shared parameterizations for the density heads, alternative constructions of $q(Z' \mid Y_{\text{true}})$, extensions to multiple sensitive attributes and structured outputs, and robustness guarantees for CI under distribution shift, with applications beyond EHR to vision, language, and tabular modalities.

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

## A  TECHNICAL DETAILS

### A.1  PROOF OF THEOREM 1

#### A.1.1  NOTATION AND ASSUMPTIONS

Let $(\Omega, \mathscr{F}, \mathbb{P})$ be an underlying probability space. We work on standard measurable spaces

$$(\mathcal{Y}_{\text{pred}}, \mathscr{Y}_p), \qquad (\mathcal{Z}, \mathscr{Z}), \qquad (\mathcal{Y}_{\text{true}}, \mathscr{Y}_t),$$

with product $\sigma$-algebras and reference (dominating) $\sigma$-finite measures $\mu_p, \mu_z, \mu_t$. Write $\mu := \mu_p \otimes \mu_z \otimes \mu_t$ and $\mu_{zt} := \mu_z \otimes \mu_t$. Random variables $Y_{\text{pred}} : \Omega \to \mathcal{Y}_{\text{pred}}$, $Z : \Omega \to \mathcal{Z}$, $Y_{\text{true}} : \Omega \to \mathcal{Y}_{\text{true}}$ are measurable.

Denote by $P_{Y_{\text{pred}}, Z, Y_{\text{true}}}$ the joint law of $(Y_{\text{pred}}, Z, Y_{\text{true}})$ and by $P_{Z, Y_{\text{true}}}$ the $(Z, Y_{\text{true}})$ marginal. Assume absolute continuity and densities:

$$p(y_p, z, y_t) = \frac{dP_{Y_{\text{pred}}, Z, Y_{\text{true}}}}{d\mu}(y_p, z, y_t), \qquad p(z, y_t) = \frac{dP_{Z, Y_{\text{true}}}}{d\mu_{zt}}(z, y_t),$$

exist, and $p(\cdot) > 0$ on its support. By disintegration, there are conditional densities $p(y_p, z \mid y_t)$, $p(y_p \mid y_t), p(z \mid y_t)$ with $p(y_p, z, y_t) = p(y_t)\, p(y_p, z \mid y_t)$, etc., for $\mu_t$-a.e. $y_t$.

Let $q(\cdot \mid y_t)$ be a Markov kernel on $(\mathcal{Z}, \mathscr{Z})$ such that $q(z \mid y_t)$ is a density (w.r.t. $\mu_z$), strictly positive for $\mu_t$-a.e. $y_t$. The auxiliary factorized law is defined as

$$Q_{Y_{\text{pred}}, Z', Y_{\text{true}}} \quad \text{with density} \quad q(y_p, z', y_t) = p(y_p, y_t)\, q(z' \mid y_t) \quad \text{w.r.t. } \mu,$$

and its $(Z', Y_{\text{true}})$-marginal $Q_{Z', Y_{\text{true}}}$ with density $q(z', y_t) = p(y_t)\, q(z' \mid y_t)$ w.r.t. $\mu_{zt}$. We also assume the KL finiteness conditions $P_{Y_{\text{pred}}, Z, Y_{\text{true}}} \ll Q_{Y_{\text{pred}}, Z', Y_{\text{true}}}$ and $P_{Z, Y_{\text{true}}} \ll Q_{Z', Y_{\text{true}}}$, so that all KL terms below are well defined.

For any probability measures $P \ll Q$ on the same measurable space, we use the convention

$$D_{\text{KL}}(P \| Q) = \int \log\left(\frac{dP}{dQ}\right) dP \in [0, \infty],$$

and the information inequality (strict convexity of $D_{\text{KL}}$): $D_{\text{KL}}(P \| Q) = 0$ iff $P = Q$ -a.s. Conditional mutual information is defined as

$$I(Y_{\text{pred}}; Z \mid Y_{\text{true}}) = \mathbb{E}_{Y_{\text{true}}}\left[ D_{\text{KL}}\big( p(Y_{\text{pred}}, Z \mid Y_{\text{true}}) \,\big\|\, p(Y_{\text{pred}} \mid Y_{\text{true}})\, p(Z \mid Y_{\text{true}}) \big) \right].$$

#### A.1.2  THEOREM AND PROOF

**Theorem 1 (Equivalence of $\xi$ and Conditional Mutual Information).** Under the assumptions above, the surrogate

$$\xi = D_{\text{KL}}\big( p(Y_{\text{pred}}, Z, Y_{\text{true}}) \,\big\|\, q(Y_{\text{pred}}, Z', Y_{\text{true}}) \big) - D_{\text{KL}}\big( p(Z, Y_{\text{true}}) \,\big\|\, q(Z', Y_{\text{true}}) \big)$$

coincides with the conditional mutual information:

$$\xi = I\big(Y_{\text{pred}}; Z \mid Y_{\text{true}}\big).$$

In particular, $\xi \geq 0$, and $\xi = 0$ if and only if $Y_{\text{pred}} \perp Z \mid Y_{\text{true}}$.

**Proof.** We write $y_p$ for $Y_{\text{pred}}$ and $y_t$ for $Y_{\text{true}}$ to lighten notation.

By the Radon–Nikodym definitions and $q(y_p, z', y_t) = p(y_p, y_t) q(z' \mid y_t)$,

$$D_{\mathrm{KL}}\big(P_{y_p,z,y_t} \,\|\, Q_{y_p,z',y_t}\big) = \int_{\mathcal{Y}_p \times \mathcal{Z} \times \mathcal{Y}_t} p(y_p, z, y_t) \, \log \frac{p(y_p, z, y_t)}{p(y_p, y_t)\, q(z \mid y_t)} \, d\mu,$$

$$D_{\mathrm{KL}}\big(P_{z,y_t} \,\|\, Q_{z',y_t}\big) = \int_{\mathcal{Z} \times \mathcal{Y}_t} p(z, y_t) \, \log \frac{p(z, y_t)}{p(y_t)\, q(z \mid y_t)} \, d\mu_{zt}. \tag{8}$$

Using Fubini/Tonelli and the identity $p(z, y_t) = \int p(y_p, z, y_t) \, d\mu_p(y_p)$, the marginal KL in (8) can be expressed over the product space (as in the main text):

$$D_{\mathrm{KL}}\big(P_{z,y_t} \,\|\, Q_{z',y_t}\big) = \int_{\mathcal{Y}_p \times \mathcal{Z} \times \mathcal{Y}_t} p(y_p, z, y_t) \, \log \frac{p(z, y_t)}{p(y_t)\, q(z \mid y_t)} \, d\mu.$$

Subtracting the two KL expressions yields

$$\xi = \int p(y_p, z, y_t) \left[ \log \frac{p(y_p, z, y_t)}{p(y_p, y_t)\, q(z \mid y_t)} \log \frac{p(z, y_t)}{p(y_t)\, q(z \mid y_t)} \right] d\mu \tag{9}$$

$$= \int p(y_p, z, y_t) \, \log \frac{p(y_p, z, y_t)\, p(y_t)}{p(y_p, y_t)\, p(z, y_t)} \, d\mu, \tag{10}$$

where the factor $q(z \mid y_t)$ cancels algebraically.

Using the chain rule $p(y_p, z, y_t) = p(y_t)\, p(y_p, z \mid y_t)$, $p(y_p, y_t) = p(y_t)\, p(y_p \mid y_t)$, $p(z, y_t) = p(y_t)\, p(z \mid y_t)$, we rewrite the logarithm as

$$\log \frac{p(y_p, z \mid y_t)}{p(y_p \mid y_t)\, p(z \mid y_t)}.$$

Therefore,

$$\xi = \int_{\mathcal{Y}_t} p(y_t) \left[ \int_{\mathcal{Y}_p \times \mathcal{Z}} p(y_p, z \mid y_t) \, \log \frac{p(y_p, z \mid y_t)}{p(y_p \mid y_t)\, p(z \mid y_t)} \, d\mu_p \, d\mu_z \right] d\mu_t(y_t).$$

By the disintegration theorem this is precisely

$$\xi = \mathbb{E}_{Y_{\mathrm{true}}} \Big[ D_{\mathrm{KL}}\big(p(Y_{\mathrm{pred}}, Z \mid Y_{\mathrm{true}}) \,\big\|\, p(Y_{\mathrm{pred}} \mid Y_{\mathrm{true}})\, p(Z \mid Y_{\mathrm{true}})\big) \Big] = I\big(Y_{\mathrm{pred}}; Z \mid Y_{\mathrm{true}}\big).$$

Each inner KL divergence is nonnegative, hence $\xi \geq 0$. Moreover, $\xi = 0$ iff $D_{\mathrm{KL}}\big(p(Y_{\mathrm{pred}}, Z \mid Y_{\mathrm{true}}) \| p(Y_{\mathrm{pred}} \mid Y_{\mathrm{true}}) p(Z \mid Y_{\mathrm{true}})\big) = 0$ almost surely in $Y_{\mathrm{true}}$, which (by strict convexity of $D_{\mathrm{KL}}$) is equivalent to $p(Y_{\mathrm{pred}}, Z \mid Y_{\mathrm{true}}) = p(Y_{\mathrm{pred}} \mid Y_{\mathrm{true}}) p(Z \mid Y_{\mathrm{true}})$ a.s., i.e., the conditional independence $Y_{\mathrm{pred}} \perp Z \mid Y_{\mathrm{true}}$. $\square$

### A.1.3 REMARKS

**Independence of the choice of** $q$   The equality $\xi = I(Y_{\mathrm{pred}}; Z \mid Y_{\mathrm{true}})$ does not depend on the specific conditional kernel $q(\cdot \mid y_t)$ (as long as the KLs are finite). The auxiliary law $q$ thus serves only as a factorized reference to make the two KL terms well defined.

**Integrability and supports.**   If $P_{Y_{\mathrm{pred}}, Z, Y_{\mathrm{true}}} \not\ll Q_{Y_{\mathrm{pred}}, Z', Y_{\mathrm{true}}}$ or $P_{Z, Y_{\mathrm{true}}} \not\ll Q_{Z', Y_{\mathrm{true}}}$, the corresponding KL divergence is $+\infty$ and the identity holds with the natural convention $\infty - \infty = I(Y_{\mathrm{pred}}; Z \mid Y_{\mathrm{true}}) = \infty$ whenever the inner conditional KL is infinite.

**On "strict convexity and separability."**   These assumptions guarantee the information inequality and equality conditions for a broad class of divergences. For $D_{\mathrm{KL}}$ they are standard and ensure that $\xi = 0$ iff the conditional factorization holds almost surely.

**Algorithm 1** FAIRNESS-AWARE I-CFM

---

**Require:** Training set $\mathcal{D} = \{(X^{(i)}, Y_{\text{true}}^{(i)}, Z^{(i)})\}_{i=1}^{N}$; fairness weight schedule $\lambda \in [0, \lambda_{\max}]$ with warm-up $E_{\text{warm}}$; EMA decay $\beta$; ODE steps $S$
**Ensure:** Generator $G_\theta$; density heads $P_\phi, Q_\psi, R_\rho, S_\chi$
 1: Initialize $\theta, \phi, \psi, \rho, \chi$; baselines $b_{\text{joint}} = b_{\text{marg}} = 0$
 2: **for** epoch $e = 1$ to $E$ **do**
 3:      $\lambda \leftarrow \lambda_{\max} \cdot \min\big(1, \frac{e}{E_{\text{warm}}}\big)$
 4:      **for** minibatch $\mathcal{B} = \{(X^{(i)}, Y_{\text{true}}^{(i)}, Z^{(i)})\}_{i=1}^{B} \subset \mathcal{D}$ **do**
 5:          Encode $h^{(i)} \leftarrow \text{Enc}(X^{(i)})$
 6:          Sample $t^{(i)} \sim \text{Unif}[0,1]$, $y_0^{(i)} \sim \mathcal{N}(0, I)$, set $y_1^{(i)} \leftarrow Y_{\text{true}}^{(i)}$
 7:          $y_t^{(i)} \leftarrow (1 - t^{(i)})y_0^{(i)} + t^{(i)}y_1^{(i)}, \quad u_t^{(i)} \leftarrow y_1^{(i)} - y_0^{(i)}$
 8:          $\mathcal{L}_{\text{CFM}} \leftarrow \frac{1}{B}\sum_{i=1}^{B}\big\|v_\theta(y_t^{(i)}, X^{(i)}, t^{(i)}) - u_t^{(i)}\big\|_2^2$
 9:          Integrate $y_0^{(i)} \rightarrow Y_{\text{pred}}^{(i)}$ with step size $1/S$ (e.g., Heun/RK2)
10:          For each label $y$, sample a permutation $\sigma_y$ on its indices; draw $\epsilon^{(i)} \sim \mathcal{N}(0, \sigma^2 I)$
11:          $Z'^{(i)} \leftarrow Z^{(\sigma_{Y_{\text{true}}^{(i)}}(i))} + \epsilon^{(i)}$
12:          $a^{(i)} \leftarrow \log P_\phi(Y_{\text{pred}}^{(i)}, Z^{(i)}, Y_{\text{true}}^{(i)})$, $b^{(i)} \leftarrow \log Q_\psi(Y_{\text{pred}}^{(i)}, Z'^{(i)}, Y_{\text{true}}^{(i)})$
13:          $c^{(i)} \leftarrow \log R_\rho(Z^{(i)}, Y_{\text{true}}^{(i)})$, $d^{(i)} \leftarrow \log S_\chi(Z'^{(i)}, Y_{\text{true}}^{(i)})$
14:          $\delta_{\text{joint}}^{(i)} \leftarrow a^{(i)} - b^{(i)}, \quad \delta_{\text{marg}}^{(i)} \leftarrow c^{(i)} - d^{(i)}$
15:          $b_{\text{joint}} \leftarrow \beta b_{\text{joint}} + (1 - \beta)\frac{1}{B}\sum_i \delta_{\text{joint}}^{(i)}$
16:          $b_{\text{marg}} \leftarrow \beta b_{\text{marg}} + (1 - \beta)\frac{1}{B}\sum_i \delta_{\text{marg}}^{(i)}$
17:          $\widehat{\xi} \leftarrow \frac{1}{B}\sum_{i=1}^{B}\big[(\delta_{\text{joint}}^{(i)} - b_{\text{joint}}) - (\delta_{\text{marg}}^{(i)} - b_{\text{marg}})\big]$
18:          $\mathcal{L}_{\text{total}} \leftarrow \mathcal{L}_{\text{CFM}} + \lambda\widehat{\xi}$
19:          Backprop through $\delta_{\text{joint}}$ to $(\theta, \phi, \psi)$; stop-grad of $\delta_{\text{marg}}$ to $\theta$; backprop $\delta_{\text{marg}}$ to $(\rho, \chi)$
20:          Update $(\theta, \phi, \psi, \rho, \chi)$ with a first-order optimizer (e.g., AdamW)
21:      **end for**
22: **end for**

---

## A.2   ALGORITHM

This subsection presents the training procedure (Algorithm 1) used throughout the paper.

**Remark**   Since the marginal term involves only $(Z, Y_{\text{true}})$ or $(Z', Y_{\text{true}})$ and carries no computational path to $\theta$, we backpropagate through $\delta_{\text{joint}}$ to update $(\theta, \phi, \psi)$ while applying stop-gradient to $\delta_{\text{marg}}$ with respect to $\theta$ (using $\delta_{\text{marg}}$ only to update $(\rho, \chi)$). Empirically, we also find that removing the dedicated marginal density heads $(\rho, \chi)$ and reusing $(\phi, \psi)$ to compute $\delta_{\text{marg}}$ achieves comparable fairness–utility trade-offs with reduced computational overhead. This simplified variant is primarily adopted in our experimental evaluation.

## B   EXPERIMENT DETAILS: SIMULATION (FOR SECTION 4)

This section provides (i) the precise simulation setup, model architectures, and training protocol used in Section 4; (ii) formal definitions of the three conditional-independence diagnostics; and (iii) additional Pareto analyses using residual HSIC and mean absolute partial correlation as fairness axes.

### B.1   DIAGNOSTICS OF CONDITIONAL INDEPENDENCE

We formalize the three diagnostics used in the simulation to assess conditional independence between predictions and sensitive attributes. Throughout, let $\{(X_i, Y_i, Z_i, G_i)\}_{i=1}^{n}$ denote i.i.d. samples, with $Y_i \in \mathbb{R}$, $Z_i \in \mathbb{R}^{d_Z}$ a sensitive covariate vector, and $G_i \in \{0, 1\}$ a binary group label. Let $\widehat{Y}_i$ be the model's prediction for $Y_i$. We write $\|\cdot\|$ for the Euclidean norm, $\text{med}\{\cdot\}$ for the median, and $\mathbb{1}\{\cdot\}$ for the indicator.

### B.1.1 EDDI$_{\text{CONT}}$: EQUALIZED DISCREPANCY VIA DISTRIBUTIONAL INVARIANCE

EDDI$_{\text{cont}}$ measures the discrepancy between the distributions of $\widehat{Y}$ across groups *conditional on* the outcome $Y$, approximated via outcome-binning.

**Binning.** Choose $K \in \mathbb{N}$ (we use $K=5$). Let $(\tau_k)_{k=0}^K$ be empirical $k/K$-quantiles of $\{Y_i\}_{i=1}^n$, and define bins $\mathcal{I}_k := \{\, i : \tau_{k-1} \leq Y_i < \tau_k \,\}$ for $k = 1, \ldots, K$.

**Within-bin MMD$^2$.** For each bin $k$, split indices by group: $\mathcal{I}_k^{(g)} := \{i \in \mathcal{I}_k : G_i = g\}$, with sizes $n_k^{(g)}$. Let $k_\sigma(a,b) = \exp(-\|a-b\|^2/(2\sigma^2))$ be a Gaussian kernel on $\mathbb{R}$ (extendable to vector $\widehat{Y}$). Set $\sigma_k$ by the (across-group pooled) median heuristic: $\sigma_k := \text{med}\{\, |\widehat{Y}_i - \widehat{Y}_j| : i \neq j, \ i,j \in \mathcal{I}_k\}$. The (biased) empirical MMD$^2$ between the two groups' predictive distributions in bin $k$ is

$$\text{MMD}_k^2 = \frac{1}{(n_k^{(0)})^2} \sum_{i,i' \in \mathcal{I}_k^{(0)}} k_{\sigma_k}(\widehat{Y}_i, \widehat{Y}_{i'}) + \frac{1}{(n_k^{(1)})^2} \sum_{j,j' \in \mathcal{I}_k^{(1)}} k_{\sigma_k}(\widehat{Y}_j, \widehat{Y}_{j'}) - \frac{2}{n_k^{(0)} n_k^{(1)}} \sum_{\substack{i \in \mathcal{I}_k^{(0)} \\ j \in \mathcal{I}_k^{(1)}}} k_{\sigma_k}(\widehat{Y}_i, \widehat{Y}_j).$$

If $n_k^{(g)} < 2$ for some $g$, we mark bin $k$ invalid and exclude it.

**Aggregation.** Let $\mathcal{K}_{\text{valid}} \subseteq \{1, \ldots, K\}$ be the set of valid bins. We report the averaged score

$$\text{EDDI}_{\text{cont}} = \frac{1}{|\mathcal{K}_{\text{valid}}|} \sum_{k \in \mathcal{K}_{\text{valid}}} \text{MMD}_k^2,$$

which is lower when $\widehat{Y}$ is (approximately) group-invariant conditional on $Y$.

### B.1.2 PARTIAL CORRELATION AFTER RESIDUALIZATION

This diagnostic estimates the linear association between $\widehat{Y}$ and each component of $Z$ after removing linear dependence on $Y$.

**Residualization.** Fit ordinary least squares (OLS) of $\widehat{Y}$ on $Y$ with intercept: $\widehat{Y}_i = a_0 + a_1 Y_i + r_i^{(\widehat{Y})}$. For each $j \in \{1, \ldots, d_Z\}$, fit OLS of $Z_{ij}$ on $Y_i$ with intercept: $Z_{ij} = b_{0j} + b_{1j} Y_i + r_i^{(Z_j)}$. Collect residuals $\mathbf{r}^{(\widehat{Y})} = (r_1^{(\widehat{Y})}, \ldots, r_n^{(\widehat{Y})})$ and $\mathbf{r}^{(Z_j)} = (r_1^{(Z_j)}, \ldots, r_n^{(Z_j)})$.

**Partial correlation.** The sample Pearson partial correlation between $\widehat{Y}$ and $Z_j$ given $Y$ equals the correlation between residuals:

$$\widehat{\rho}_j = \frac{\sum_{i=1}^n \left(r_i^{(\widehat{Y})} - \overline{r}^{(\widehat{Y})}\right)\left(r_i^{(Z_j)} - \overline{r}^{(Z_j)}\right)}{\sqrt{\sum_{i=1}^n \left(r_i^{(\widehat{Y})} - \overline{r}^{(\widehat{Y})}\right)^2} \sqrt{\sum_{i=1}^n \left(r_i^{(Z_j)} - \overline{r}^{(Z_j)}\right)^2}},$$

where bars denote sample means. We summarize by the mean absolute partial correlation

$$|\rho|_{\text{partial}} = \frac{1}{d_Z} \sum_{j=1}^{d_Z} |\widehat{\rho}_j|.$$

Smaller values indicate weaker linear dependence of predictions on sensitive covariates after accounting for $Y$.

### B.1.3 RESIDUAL HSIC

HSIC detects (potentially nonlinear) dependence between $\widehat{Y}$ and $Z$ after residualization with respect to $Y$.

**Residuals.** Use the same residuals $r_i^{(\widehat{Y})}$ and $r_i^{(Z_j)}$ as in Appendix B.1.2.

**Kernels.** For each $j$, define Gaussian kernels $k_{\sigma_{\widehat{Y}}}(a, b) = \exp(-|a - b|^2/(2\sigma_{\widehat{Y}}^2))$ on $\mathbb{R}$ and $\ell_{\sigma_j}(a, b) = \exp(-|a - b|^2/(2\sigma_j^2))$ on $\mathbb{R}$. Choose bandwidths by the median heuristic: $\sigma_{\widehat{Y}} = \mathrm{med}\{|r_i^{(\widehat{Y})} - r_{i'}^{(\widehat{Y})}| : i \neq i'\}$ and $\sigma_j = \mathrm{med}\{|r_i^{(Z_j)} - r_{i'}^{(Z_j)}| : i \neq i'\}$.

**Empirical HSIC (biased).** Let $K \in \mathbb{R}^{n \times n}$ with $K_{ii'} = k_{\sigma_{\widehat{Y}}}(r_i^{(\widehat{Y})}, r_{i'}^{(\widehat{Y})})$, $L^{(j)} \in \mathbb{R}^{n \times n}$ with $L_{ii'}^{(j)} = \ell_{\sigma_j}(r_i^{(Z_j)}, r_{i'}^{(Z_j)})$, and $H = I_n - \frac{1}{n}\mathbf{1}\mathbf{1}^\top$. The (biased) HSIC estimate between the residuals is

$$\mathrm{HSIC}_j = \frac{1}{n^2}\mathrm{tr}(KHL^{(j)}H).$$

We report the mean residual HSIC across coordinates

$$\mathrm{HSIC}_{\text{partial}} = \frac{1}{d_Z}\sum_{j=1}^{d_Z}\mathrm{HSIC}_j.$$

Lower scores indicate weaker (possibly nonlinear) dependence between predictions and sensitive attributes after conditioning on $Y$.

### B.2 SIMULATION SETUP AND TRAINING PROTOCOL

**Synthetic data and splits.** We generate $n$ i.i.d. samples $\{(X^{(i)}, Y^{(i)}, Z^{(i)}, G^{(i)})\}_{i=1}^n$ as described in Section 4 with

$$T = 48, \quad F = 48, \quad F_1 = 1, \quad F_2 = 5,$$

and split into $n_{\text{train}} = 2000$ and $n_{\text{test}} = 3000$ with fixed seeds for reproducibility. Group effects and couplings follow the simulation kernel in Section 4: a group strength on $Y$ ("y_group_strength"= 0.8), a direct group separation in $Z$ ("z_group_strength"= 1.0), and a mediated $Y \to Z$ coupling ("alpha_y2z"= 1.0). All reported curves average across multiple independent replicates (seeds), as specified below.

**Temporal encoder (TemporalU-Net).** We encode $X \in \mathbb{R}^{T \times F}$ with a 1D U-Net operating along time (input layout to the conv stack is (batch, $F$, $T$)):

- **Down path (encoder):** four levels with channel widths $[C_0, 2C_0, 4C_0, 8C_0]$ where $C_0$=*unet_channels*=52. Each level uses two Conv1d(ker=3, pad= 1)$\to$BatchNorm1d$\to$ReLU blocks, then MaxPool1d(2).

- **Bottleneck:** Conv1d blocks as above at width $16C_0$.

- **Up path (decoder):** ConvTranspose1d(ker=2, stride= 2) upsampling; skip connections are concatenated channel-wise with encoder features.

- **Head and pooling:** a $1 \times 1$ Conv1d maps to *output_dim= hidden_dim*, followed by global average pooling over time, yielding $h \in \mathbb{R}^{hidden\_dim}$. A small alignment utility crops/pads when upsampled lengths differ by 1 (due to pooling).

**Vector field $v_\theta(y_t, X, t)$.** Given $y_t \in \mathbb{R}^{F_1}$, the encoder feature $h \in \mathbb{R}^{128}$, and *a scalar* time input $t \in [0, 1]$ (no sinusoidal embedding), we concatenate $[y_t; h; t]$ and feed a depth-$L$=4 MLP:

MLP widths $[F_1 + 128 + 1, 128, 128, 1]$ with ReLU and Dropout(0.1) between hidden layers,

This parameterizes $\frac{dy}{dt} = v_\theta(y_t, X, t)$ with $y_0 \sim \mathcal{N}(0, I)$. ODE integration in simulation uses *forward Euler* with $S$=100 steps by default.

**Density heads for $\widehat{\xi}$.** To instantiate the surrogate CI penalty with *normalized* densities, we model each required law by a continuous-time flow trained via CFM. Concretely, we learn two heads

$$P_\phi : (y_{\text{pred}}, z, y_{\text{true}}) \mapsto \log p_\phi, \quad Q_\psi : (y_{\text{pred}}, z', y_{\text{true}}) \mapsto \log q_\psi,$$

where each head parameterizes a time-dependent vector field $v_\bullet(\zeta, t)$ that transports a simple base $p_0(\zeta_0)$ (e.g., standard Gaussian) to the target density at $t=1$. The log-density is obtained by the continuous change-of-variables identity

$$\log p_T(\zeta) \;=\; \log p_0(\zeta_0) \;-\; \int_0^T \nabla_\zeta \cdot v_\bullet\big(\phi_{t\leftarrow 0}(\zeta_0), t\big)\, dt, \qquad \zeta_0 = \phi_{0\leftarrow T}(\zeta),$$

with the divergence $\nabla_\zeta \cdot v_\bullet$ estimated by Hutchinson probes (typically $m \in \{1, 2\}$). Each $v_\bullet$ is implemented as a small MLP (two hidden layers, width 256, ReLU; dropout 0.1) that consumes $\zeta$ concatenated time and (when applicable) context $y_{\text{true}}$. We train the two flows with the standard CFM regression loss on linear interpolants, which avoids inner ODE solves during training while yielding calibrated log-densities at evaluation time.

At training time, $\widehat{\xi}$ is computed from the *normalized* log-densities

$$\log P_\phi, \; \log Q_\psi$$

in the KL–difference estimator. This CFM instantiation enables likelihood auditing and removes any reliance on unnormalized energy surrogates.

**CI surrogate and factorized samples.** We form $z'^{(i)} = z^{(\sigma(i))} + \epsilon$ with a within-minibatch random permutation $\sigma$ and $\epsilon \sim \mathcal{N}(0, \sigma_{\text{noise}}^2 I)$ ($\sigma_{\text{noise}}{=}0.05$), and use an EMA baseline ($\beta{=}0.99$) for variance reduction in $\widehat{\xi}$. The total objective is $\mathcal{L}_{\text{total}} = \mathcal{L}_{\text{CFM}} + \lambda \widehat{\xi}$, with a linear warm-up of $\lambda$ over the first 20% of epochs.

**Training and evaluation.** We optimize with AdamW (lr $10^{-3}$, weight decay $10^{-4}$), batch size 64, and 50 epochs. We sweep $\lambda \in \{0, 0.01, 0.05, 0.1\}$, average over multiple random seeds, and report $-$RMSE (higher is better) versus CI metrics (lower is better: $\text{EDDI}_{\text{cont}}$, $\text{HSIC}_{\text{partial}}$, and $|\rho|_{\text{partial}}$). Data are provided to the U-Net as $(\text{batch}, F, T)$; if arrays are $(\text{batch}, T, F)$ we transpose before encoding

### B.3    Additional Results: Pareto Fronts with HSIC and Partial Correlation

We complement the main-text Pareto analysis with two figures that sweep the fairness weight $\lambda$ and plot performance against (i) residual HSIC and (ii) mean absolute partial correlation. Curves show the mean across 10 random seeds. All other hyperparameters match Section 4.

**Findings.** In both figures, increasing $\lambda$ produces monotone leftward movement (stronger conditional independence) with minimal vertical drift (near-constant RMSE), yielding smooth Pareto fronts. Improvements saturate beyond $\lambda \approx 0.1$, mirroring the $\text{EDDI}_{\text{cont}}$ curves in the main text. The HSIC- and $|\rho|_{\text{partial}}$-based fronts are qualitatively consistent, supporting that the fairness regularizer reduces both nonlinear (HSIC) and linear (partial correlation) residual dependencies with negligible accuracy loss.

## C    Experiment Details: Real-data (for Section 5)

**Summary of datasets.** We evaluate on two large-scale, real-world EHR datasets: MIMIC-III (Johnson et al., 2016) and MIMIC-IV (Johnson et al., 2023). Both datasets contain de-identified health records of patients admitted to intensive care units (ICUs) or emergency departments at Beth Israel Deaconess Medical Center (BIDMC), comprising multivariate time-series data (e.g., vital signs, lab tests) and demographic information.

Following established preprocessing protocols (Harutyunyan et al., 2019; Wang et al., 2024), we extract 26 continuous-valued clinical variables from MIMIC-III and 25 from MIMIC-IV, sampled hourly during the first 48 hours of ICU admission. Table 3 summarizes dataset statistics: we obtain 18,143 ICU stays from MIMIC-III and 21,773 from MIMIC-IV, and randomly split each dataset into training, validation, and test sets using a 7:1:2 ratio.

We extract five categories of continuous-valued clinical predictors from the MIMIC-III and MIMIC-IV datasets: vital signs, blood gases, renal function, metabolic panel, and hematology. We focus

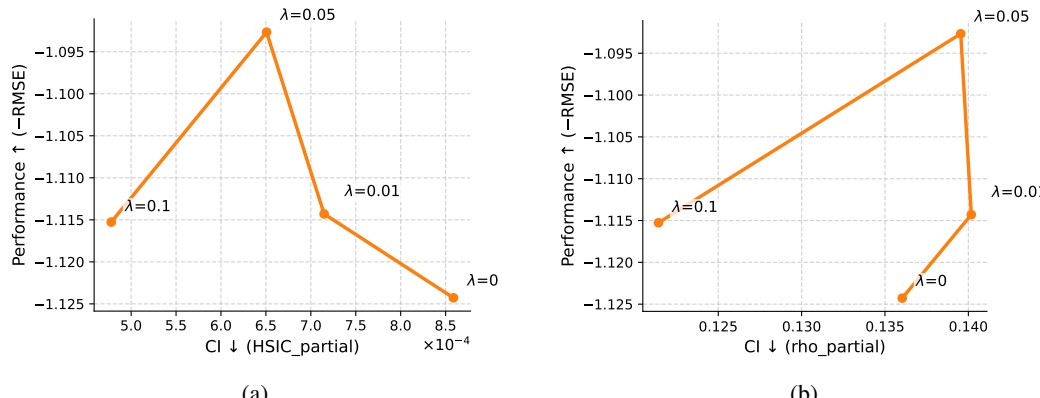

(a)                                                                                    (b)

Figure 4: **Pareto frontiers for independence metrics.** (a) Performance ($-$RMSE; higher is better) versus $\text{HSIC}_{\text{partial}}$ (lower is better) across $\lambda$. Larger $\lambda$ shifts solutions leftward (stronger CI) with minor vertical change, exhibiting a clear "elbow" near $\lambda \approx 0.1$. (b) Performance ($-$RMSE; higher is better) versus $|\rho|_{\text{partial}}$ (lower is better) across $\lambda$. Trends closely match (a), indicating consistent CI gains under linear and nonlinear diagnostics.

Table 3: Summary statistics of the datasets used.

| Dataset | MIMIC-III | | | MIMIC-IV | | |
|---|---|---|---|---|---|---|
| Split | Training | Validation | Test | Training | Validation | Test |
| Total | 12,672 | 1,833 | 3,638 | 15,112 | 2,188 | 4,473 |
| Missing rate | 72.81% | 72.71% | 72.73% | 71.16% | 71.17% | 71.23% |
| Positive (**IHM**) | 1,481 | 237 | 446 | 1,702 | 243 | 511 |
| Positive (**READM**) | 2,268 | 364 | 647 | 2,648 | 390 | 814 |

exclusively on the first 48 hours of patient data recorded after ICU admission, sampling observations at hourly intervals. Admissions with fewer than 48 hours of recorded data are excluded. Detailed predictor information is summarized in Table 4. We preprocess the raw data using the pipeline proposed by Harutyunyan et al. (2019). Notably, due to the extreme sparsity of arterial oxygen pressure data in MIMIC-IV, we include this predictor only for the MIMIC-III dataset. Consequently, the total number of predictors is 26 for MIMIC-III and 25 for MIMIC-IV.

Table 4: Summary of clinical predictors in longitudinal data for MIMIC-III/IV datasets, * indicates the predictor is only available in MIMIC-III.

| Category | Predictors |
|---|---|
| Vital Signs | Heart Rate, Systolic Blood Pressure, Diastolic Blood Pressure, Mean Blood Pressure, Respiratory Rate, Body Temperature, Oxygen Saturation |
| Blood Gases | Arterial Base Excess, Arterial Carbon Dioxide Pressure, Arterial Oxygen Pressure*, Arterial pH |
| Renal Function | Blood Urea Nitrogen, Creatinine |
| Metabolic Panel | Ionized Calcium, Serum Chloride, Serum Glucose, Fingerstick Glucose, Anion Gap, Serum Bicarbonate, Magnesium, Serum Potassium, Serum Sodium |
| Hematology | Serum Hematocrit, Hemoglobin, Platelet Count, White Blood Cell Count |

We also extract five sensitive attributes from the MIMIC-III and MIMIC-IV datasets: insurance type, marital status, race, gender, and age. Each attribute contains several subgroups, whose compositions vary between the two datasets. For instance, the insurance attribute comprises five subgroups (*Medicare, Medicaid, Government, Self Pay, and Private*) in MIMIC-III, whereas it includes only

three subgroups (*Medicare, Medicaid, and Other*) in MIMIC-IV due to differences in recording standards.

**Implementation Details and Hyperparameters.**   Our method is implemented in Python 3.11 using *PyTorch* 2.0. All models are trained for a maximum of 100 epochs, and the best-performing model is selected based on AUROC on the validation set. The final performance is reported on the test set. To calculate the F1 score and fairness metrics, we select thresholds corresponding to the best F1 score obtained on the validation set. We utilize the *Adam* optimizer and implement early stopping if no improvement in validation AUROC is observed for 10 consecutive epochs to prevent overfitting. For adversarial fine-tuning, we initiate advesarial low-rank fine-tuning after training the MINE module for 30 epochs. For all baselines, we concatenate time series data and its missing mask(e.g., 0 for observed data and 1 for missing data) as the input. All experiments are conducted using a single NVIDIA RTX-4090 GPU with a batch size of 128. Hyperparameter tuning is performed using grid search on the validation set, with the following search spaces:

- Dropout ratio: $\{0, 0.1, 0.2, 0.3\}$
- Learning rate: $\{1 \times 10^{-4}, 5 \times 10^{-5}, 1 \times 10^{-5}\}$
- Mutual information regularization coefficient $\lambda_{\text{MI}}$: $\{0.1, 0.2, 0.5\}$

We report results from the optimal hyperparameter settings identified through validation.

**Detailed Descriptions of Baseline Methods.**

- **CNN** (LeCun et al., 1998): Convolutional Neural Networks utilize convolutional layers to automatically extract hierarchical representations, enabling the learning of complex decision boundaries for predictive tasks.
- **RNN** (Elman, 1990): Recurrent Neural Networks process sequential data by recursively passing hidden states through time steps, making them effective for modeling temporal dependencies.
- **LSTM** (Graves & Graves, 2012): Long Short-Term Memory networks are specialized recurrent architectures designed to effectively capture long-term dependencies and mitigate the vanishing gradient problem inherent in standard RNNs.
- **Transformer** (Vaswani et al., 2017): Transformer architectures leverage self-attention mechanisms, allowing models to efficiently capture global dependencies without recurrent connections, thus demonstrating excellent generalization across multiple domains.
- **FFVAE** (Creager et al., 2019): It employs adversarial decorrelation within a variational autoencoder framework to disentangle sensitive attributes from latent representations, ensuring fairness in downstream predictions.
- **FarconVAE** (Oh et al., 2022): It combines variational autoencoder techniques with contrastive learning objectives to achieve fair representation learning through disentanglement.
- **FairEHR-CLP** (Wang et al., 2024): This model integrates generative adversarial networks for synthesizing counterfactual patient data, followed by contrastive learning to explicitly reduce prediction biases across demographic groups.
- **FLMD** (Liu et al., 2023): It employs deconfounder theory to infer and incorporate latent confounders, improving fairness by addressing unobserved biases within the dataset.

