# OpenReview forum: "Fair Generation via Conditional-Independent Flow Matching"
_ICLR.cc/2026/Conference — ICLR 2026 Conference Withdrawn Submission_

### Official Review · Reviewer_TyrU · 2025-10-15

**Soundness:** 2
**Presentation:** 2
**Contribution:** 2
**Rating:** 4
**Confidence:** 3

**Summary:**

This paper introduces CI-CFM, a novel framework that integrates Conditional Flow Matching (CFM) with a density-based regularizer to enforce fairness in predictive models. The theoretical contribution, particularly Theorem 1 which establishes an equivalence between the proposed divergence-difference objective and Conditional Mutual Information (CMI), is a significant strength. However, the paper could be substantially improved by addressing concerns regarding its framing, the scope of its experimental validation, and the articulation of its core technical novelty.

**Strengths:**

1. Proposed limitations are common issues and important in fairness studies.

2. Adopting CFM for enforcing fair predictions is intriguing.

**Weaknesses:**

1. The Title/scope mismatch (“fair generation” vs. classifier focus). Although the method is framed as “generation,” all core tasks are conditional prediction/classification (IHM, readmission). The title and Intro speak broadly about fairness in generative models, but the proposed method and experiments focus on general classifications instead. This is quite confusing.

2. The experiments are confined to the EHR domain (MIMIC-III/IV). While this is a valuable application area, fairness is a broad field with established benchmarks. The lack of evaluation on standard fair classification datasets (e.g., Adult, COMPAS, CelebA) makes it difficult to assess the method's generalizability and compare its performance against a wider array of fairness algorithms.

3. Some parts are hard to follow. For example, EHR needs more explanation. The mentioned limitations of current studies need further justification/explanation.

4. Related work: “Fairness-Aware Generation” vs. fair classifiers. The section title suggests generation, but the cited papers are essentially fair representation/classifier methods.

5. The core approach combines an existing architecture (CFM) with established concepts from fairness literature. Specifically, the technique of creating a factorized distribution by permuting or injecting noise into sensitive attributes to enforce independence is a well-known strategy. From this perspective, the contribution might appear straightforward.

6. While the paper mentions that CFM offers stable training, it needs to more explicitly argue why this stability is particularly crucial for enforcing fairness compared to other methods (e.g., GANs or VAEs), perhaps with empirical evidence like loss curve comparisons. The contribution appears to be an application of CFM to a known fairness problem rather than a fundamental extension of CFM itself.

**Questions:**

1. Intro mixes LLMs/generative models with clinical predictors, which is confusing. Authors should further clarify their focus.

2. Intro’s heavy EHR focus reads as if the proposed method is for EHR-only. So, are the proposed methods valid with general classification datasets?

---

### Official Review · Reviewer_iXJY · 2025-10-31

**Soundness:** 2
**Presentation:** 2
**Contribution:** 2
**Rating:** 2
**Confidence:** 3

**Summary:**

The authors propose a new technique called **CI-CFM** which employs multiple heads to implement a regularizer for enforcing conditional independence constraints for group fairness to effectively use a conditional flow-matching backbone to generate the predictions. They report results on healthcare datasets, performing significantly better than the baselines and provide appropriate theoretical justification for their algorithm. They also provide the required code.

**Strengths:**

- **[S1]** The empirical results seem quite strong and well written. I’m pleased by the overall format of reporting of quantitative data and the comparisons across a wide range of baselines. Although the 2 datasets tested are very similar in domain, I think reducing the scope of the method to highlight the focus on healthcare data could alleviate those concerns.

- **[S2]** The research question the authors aim to address is of great importance, and there seems to be a natural marriage in trying to use conditional flow matching for the conditional independence task for fairness, and having a robust and efficient method to do the same would be a great step for the community.

- **[S3]** The authors provide open source code which seems to be well formatted and clean. The code seems complete and coherent, seemingly implementing the algorithm the authors propose.

**Weaknesses:**

- **[W1]** There seems to be a mismatch in the theoretical and empirical method discussed in the manuscript. The description in Section 3.4 using four neural density estimators to calculate the surrogate divergence does not match Appendix A.2 where the marginal density heads are dropped without justification. There are no ablations to highlight the comparable fairness-utility tradeoffs as claimed by the remark on Line 791. This seems to be misleading.

- **[W2]** The claim in Theorem 1 seems a bit strong and hard to digest with the assumptions given that it’s difficult to show that the estimators truly train to convergence and the practical estimate is accurate.

- **[W3]** Line 280 talks about the calculation of the computationally expensive integral of the divergence, which authors claim can perhaps be simplified by using estimators. But as the authors admit to memory overheads and computational increase in Line 332. Comparisons with existing methods, particularly the adversarial ones, on the computation and memory costs could help alleviate these concerns. It is still unclear to me how the integral and the divergence is calculated empirically and how accurate and computationally expensive the terms are.

- **[W4]** Line 159 talks about building on top of Ahuja et al. (2021) but it is not clear what the significant differences are, and more importantly, there are no comparisons to any of the adversarial CI enforcement methods which are in spirit closest to the proposed methodology.

- **[W5]** The clarity of tables could be improved, it’s very hard to read and infer anything from Table 2 due to the compactness and density of information. Could help to split the tables, remove citations from the names and report certain metrics in the appendix.

**Questions:**

- **[Q1]** It’s unclear to me why in Figure 2, the RMSE and CI both improve on moving leftwards from $\lambda= 0.01$ to $\lambda=0.05$, some ablations on more fine-grained values of $\lambda$ could help understand the pareto frontiers better.

- **[Q2]** See [W3] for context. The authors say they use either automatic differentiation or stochastic trace estimators but do not elaborate how they make this decision, is it situation based, parameter based, or exactly how?

- **[Q3]** See [W4] for context. How does the proposed algorithm do, compared to Ahuja et al. (2021) or even Madras et al. (2018) across the accuracy-fairness tradeoff?

- **[Q4]** See [W1] and [W2] for context. Could you help bridge the gap between theory and practice and help map the theoretical analysis of the actual final algorithm and architecture that was implemented and the results were generated on, or provide the exact outcomes of the runs with the precise algorithm as described in the theory?

**Details Of Ethics Concerns:**

N.A.

---

### Official Review · Reviewer_JsT2 · 2025-10-31

**Soundness:** 3
**Presentation:** 2
**Contribution:** 2
**Rating:** 4
**Confidence:** 4

**Summary:**

This paper proposed a flow matching method for enforcing equalized odds (conditional independence of $\hat Y$ and $Z$ given $Y$), by minimizing the mutual information. Such method yields a non-adversarial optimization framework and few step sampling. The authors also provide simulation and empirical studies to confirm the performance.

**Strengths:**

1. The method has a sound theoretical justification and motivation.
2. The paper is well structured and complete with sound theoretical insights and empirical experiments.

**Weaknesses:**

1. Fairness metrics: the paper is essentially targeted at the fairness metric "equalized odds" which was proposed by https://arxiv.org/abs/1610.02413. However, I didn't seem to see a discussion about this metric and the comparison with other metrics (equal opportunity/demographic parity etc.). I would be better if there were a short literature review as to why the authors targeted this specific metric instead of others.
2. Problem settings: It wasn't so clear what kind of learning problem the proposed method should be applied. Specifically, is this method only developed for multivariate time-series (spatial-temporal data)? What kind of constraint should the required input $(X, Z, Y)$ have? It would be better if the authors could include a specification on the type of problem that their method targets.
3. Baselines/literature review: Although the authors mentioned some of the proposed method for equalized odds (kernel penalty, adversarial training), it still lacks a more detailed comparison as to what category of methods the proposed one falls into (following the previous point), e.g., for tabular data there've been numerous methods proposed for equalized odds, [1] for minimizing HGR coefficient, [2] for minimizing mutal information (strongly related to the paper). [3] using adversarial sampling. It will be of vital importance if the authors clarify more about the difference between the paper and others in terms of problem setting and methods used.
3. Experiments: I appreciate that the authors have both simulation and real-world dataset evaluation. However, I feel like the real-world dataset part is rather limited, following up on the previous points (again), it only considers the dataset that's in one specific field and specific data inputs (multivariate timeseries). And the baselines used are not extensively discussed in the related work section, which indeed caused a little bit of confusion.
4. "High dimension challenge": the paper claimed "Existing approaches—kernel penalties, adversarial debiasing, and mutual-information bounds—often scale poorly in high dimensions, are unstable to train, or lack auditable likelihoods". However, it was not so clear as to what "high dim" means in this problem setting, are they refering to the dimension of $X$ or $Z$? For the dimensionality in $Z$, there've also been some literature discussing this: [4] discusses the equal opportunity (which is an approximation of equalized odds in the binary classification with binary $Z$s) for multiple subgroups, [5] discusses the equalized odds for multivariate $Z$ for tabular data with conditional permutation etc. In terms of approaching the "high-dim" problem, how does the paper compare with the previous literature? Again, an extensive comparative analysis seems needed.


[1]  Mary, Jérémie, Clément Calauzenes, and Noureddine El Karoui. "Fairness-aware learning for continuous attributes and treatments." International conference on machine learning. PMLR, 2019.

[2] Cho, Jaewoong, Gyeongjo Hwang, and Changho Suh. "A fair classifier using mutual information." 2020 IEEE international symposium on information theory (ISIT). IEEE, 2020.

[3] Romano, Yaniv, Stephen Bates, and Emmanuel Candes. "Achieving equalized odds by resampling sensitive attributes." Advances in neural information processing systems 33 (2020): 361-371.

[4] Kearns, Michael, et al. "Preventing fairness gerrymandering: Auditing and learning for subgroup fairness." International conference on machine learning. PMLR, 2018.

[5] Lai, Yuheng, Leying Guan. "FairICP: Encouraging Equalized Odds via Inverse Conditional Permutation." International conference on machine learning. PMLR, 2025.

**Questions:**

Please see weaknesses.

---

### Official Review · Reviewer_bQpk · 2025-11-01

**Soundness:** 3
**Presentation:** 3
**Contribution:** 3
**Rating:** 4
**Confidence:** 3

**Summary:**

This paper proposes CI-CFM, a fairness-aware generative framework that combines conditional flow matching (CFM) with a density-based regularizer to ensure predictions are independent of sensitive attributes (e.g., race, gender) when given true labels. Its core strength is proving the framework’s key objective (a divergence-difference measure) equals conditional mutual information, providing a rigorous basis for fairness. Experiments on synthetic data and MIMIC-III/IV medical records show it balances accuracy and fairness well.

**Strengths:**

1. Novelty. Unlike prior work relying on approximate fairness surrogates (e.g., kernel penalties, adversarial training), CI-CFM’s key objective directly maps to conditional mutual information under mild conditions. This makes fairness constraints interpretable and auditable—critical for high-stakes fields like healthcare.
2. CI-CFM fits medical records well: its temporal encoder captures time-dependent patterns in irregular clinical data, few-step sampling enables real-world use, and density-based estimation handles EHR sparsity better than contrastive/adversarial methods. It improves accuracy (e.g., 0.05-0.50 AUROC gain) and reduces bias (e.g., 0.07-0.32 EDDI drop) on MIMIC datasets.
3. Experiments. The paper validates performance on both synthetic (regression) and real-world (medical classification) tasks, using multiple fairness (EDDI, equalized odds) and accuracy metrics. 95% confidence intervals enhance result reliability.
4. Single-stage, non-adversarial training avoids the instability of min-max optimization. A variance-reduction technique for the fairness objective and clear complexity analysis (3-5x compute, 2-3x memory vs. vanilla CFM) highlight practicality.

**Weaknesses:**

1. Missing Technical Details. The paper mentions four density modules critical to fairness but provides no details on their architecture (e.g., network depth/width) or training (e.g., learning rate, convergence checks). The "20% epoch warm-up" for the fairness weight lacks justification or validation of its impact on training.
2. Incomplete Experimental Design. Key components (temporal encoder type, ODE step count, variance-reduction technique) are not ablated to show their necessity. Baseline models (e.g., FairEHR-CLP) lack implementation details (e.g., same EHR preprocessing as CI-CFM) for fair comparison. Sensitive attributes are analyzed as a group, with no breakdown of performance on individual attributes (e.g., bias reduction for race vs. gender).

**Questions:**

see weakness

---

### Note · Authors · 2025-11-19

I have read and agree with the venue's withdrawal policy on behalf of myself and my co-authors.